# The Pre-Stroke Induction and Normalization of Insulin Resistance Respectively Worsens and Improves Functional Recovery

**DOI:** 10.3390/ijms24043989

**Published:** 2023-02-16

**Authors:** Ellen Vercalsteren, Dimitra Karampatsi, Doortje Dekens, Aikaterini Letsiou, Alexander Zabala, Mihaela Romanitan, Thomas Klein, Thomas Nyström, Vladimer Darsalia, Cesare Patrone

**Affiliations:** 1NeuroCardioMetabol Group, Department of Clinical Science and Education, Södersjukhuset, Internal Medicine, Karolinska Institutet, 118 83 Stockholm, Sweden; 2Neurology Department, Internal Medicine, Södersjukhuset, 118 83 Stockholm, Sweden; 3Boehringer Ingelheim Pharma GmbH & Co., KG, 88400 Biberach, Germany

**Keywords:** type 2 diabetes, stroke, insulin resistance, neurological recovery, Rosiglitazone

## Abstract

Type 2 diabetes (T2D) impairs post-stroke recovery, and the underlying mechanisms are unknown. Insulin resistance (IR), a T2D hallmark that is also closely linked to aging, has been associated with impaired post-stroke recovery. However, whether IR worsens stroke recovery is unknown. We addressed this question in mouse models where early IR, with or without hyperglycemia, was induced by chronic high-fat diet feeding or sucrose supplementation in the drinking water, respectively. Furthermore, we used 10-month-old mice, spontaneously developing IR but not hyperglycemia, where IR was normalized pharmacologically pre-stroke with Rosiglitazone. Stroke was induced by transient middle cerebral artery occlusion and recovery was assessed by sensorimotor tests. Neuronal survival, neuroinflammation and the density of striatal cholinergic interneurons were also assessed by immunohistochemistry/quantitative microscopy. Pre-stroke induction and normalization of IR, respectively, worsened and improved post-stroke neurological recovery. Moreover, our data indicate a potential association of this impaired recovery with exacerbated neuroinflammation and a decreased density of striatal cholinergic interneurons. The global diabetes epidemic and population aging are dramatically increasing the percentage of people in need of post-stroke treatment/care. Our results suggest that future clinical studies should target pre-stroke IR to reduce stroke sequelae in both diabetics and elderly people with prediabetes.

## 1. Introduction

Type 2 diabetes (T2D), a metabolic disease characterized by hyperglycemia and insulin resistance (IR), is an expanding global health issue [1]. T2D is a strong risk factor for stroke [2] and the leading cause of mortality from cardiovascular disease [3]. T2D also worsens stroke recovery [4] and increases the disability burden, being a strong predictor of post-stroke dependency on assistive care [5,6,7,8]. Despite the availability of pharmacological and lifestyle change strategies that reduce stroke risk in T2D [3], efficacious therapies to improve post-stroke recovery and reduce stroke sequelae in this group of people are entirely lacking.

Hyperglycemia is one of the hallmarks of T2D. In stroke patients (with or without diabetes), hyperglycemia at admission is associated with a poor outcome [9,10,11] and, to date, interventional studies have been mainly focused on controlling hyperglycemia acutely after hospitalization. However, the benefit of treating hyperglycemia after stroke remains elusive, and intensive glucose control has shown no additional clinical benefit compared to standard in-hospital glycemic control [12], suggesting that other underlying pathophysiological mechanisms of diabetes may be responsible for poor stroke recovery. IR is another major T2D hallmark, which is also an independent risk factor for cardiovascular disease [13,14], including stroke [15]. Similar to hyperglycemia, IR has been associated with poor stroke outcomes [14,16]. Moreover, we recently showed that a dietary change leading to weight loss before stroke improved stroke recovery and this effect occurred in association with the pre-stroke normalization of IR [17], pointing towards the detrimental role of IR in impairing stroke recovery in T2D. However, the causative role of IR in impaired stroke recovery has not yet been established, and interventional studies aiming to normalize IR pre-stroke in the preventive perspective to limit stroke sequelae in T2D individuals are lacking. Determining whether the pre-stroke normalization of IR improves stroke recovery could have strong implications beyond T2D. Indeed, there is a high prevalence of IR in the elderly (prediabetes) [18,19,20], who have the highest stroke mortality and more severe and lasting stroke sequelae [21,22].

The T2D-induced cellular mechanisms of neurovascular damage and repair that are responsible for impaired stroke recovery are also largely uncharacterized. Persistent hyperglycemia [11] and IR [21,23,24] have been shown to be predictors of post-stroke severity and poor functional outcomes. However, it is essentially impossible to individually determine the effects of these conditions on stroke clinically since both are present in T2D patients. Furthermore, the detrimental effects of T2D on some self-repairing mechanisms during the stroke recovery phase, e.g., neuroplasticity [25,26,27], stroke-induced neurogenesis [25,27], stroke-induced oligodendrogenesis and white matter repair [27,28], vascular function [29,30] (also reviewed in [31]) and neuroinflammation [17,32,33,34] (reviewed in [35]), have been recently demonstrated. However, the specific effects of hyperglycemia and IR on these cellular mechanisms are essentially unknown due to the very close interplay of these two factors during the diabetic disease. Finally, another aspect that is poorly understood is whether the impairment of stroke recovery by T2D is mainly determined by compromised neuronal function already before stroke or whether the negative effects of T2D interfere in the reparative and neuroplasticity processes that occur after stroke.

In an attempt to understand the effects of IR on stroke recovery, independently of hyperglycemia, we took advantage of mouse models of early IR/decreased insulin sensitivity with or without concurrent hyperglycemia. Specifically, the aim of this study was to determine whether IR before stroke worsens post-stroke functional recovery. Furthermore, we investigated the potential association between impaired functional recovery induced by IR and stroke-induced brain damage and cellular mechanisms of neural repair. Finally, we assessed whether the pre-stroke pharmacological targeting of IR improves functional recovery.

## 2. Results

### 2.1. HFD Induces IR and Hyperglycemia while 30% Sucrose in Drinking Water Leads to Early IR without Hyperglycemia

To induce IR with (IR-hyperglycaemic (IR-HG), Study 1) or without (IR-normoglycaemic (IR-NG), Study 2) hyperglycemia, mice were exposed to 4 months of high-fat diet (HFD) or 30% sucrose in drinking water, respectively (see Materials and Methods and Figure 1a,b).

Four months of exposure to HFD (IR-HG mice) induced obesity (Figure 2a), hyperglycemia (Figure 2b), hyperinsulinemia (Figure 2c) and increased serum plasminogen activator inhibitor 1 (PAI-1) levels (Figure 2d), which contribute to the worsening of the (hypo)fibrinolytic state of diabetic patients [36] and are also associated with major adverse cardiovascular events (MACE) [37]. Moreover, the response to insulin was significantly diminished in IR-HG mice compared to CTRL mice, as detected by two-way repeated-measures ANOVA (diet effect: *p* = 0.0018) and the calculated area under the curve (Figure 2e,f). In contrast to HFD, which induced a weight gain of 23% over controls, exposure to 30% sucrose in drinking water for four months (IR-NG mice) only resulted in a 7% weight gain (Figure 2g) and, importantly, did not induce hyperglycemia (Figure 2h). However, the exposure to sucrose-supplemented drinking water did result in hyperinsulinemia (Figure 2i), elevated plasma PAI-1 levels (Figure 2j) and decreased insulin sensitivity, detected by two-way repeated-measures ANOVA (drinking water effect: *p* = 0.0004) and the calculated area under the curve (Figure 2k,l). These results suggest that HFD mimics more pronounced T2D features versus 30% sucrose in the drinking water.

### 2.2. IR with or without Hyperglycemia Similarly Impairs Stroke Recovery

After inducing stroke experimentally, the IR-HG mice showed a significant impairment in the recovery of grip strength (Figure 2m) and stroke-induced sensory lateralization (corridor test) versus CTRL mice (Figure 2n). Two-way repeated-measures ANOVA analyses revealed a significant interaction between treatment (HFD exposure) and time (recovery) (grip strength test *p* = 0.026, corridor task *p* = 0.003). Cerebral blood flow reduction during MCAO and neuronal loss in the striatum were not different between the groups (*p* = 0.236) (Figure 2o).

Early IR induced by 30% sucrose in IR-NG mice also led to the impaired recovery of grip strength (Figure 2p) and stroke-induced sensory lateralization versus CTRL (Figure 2q), as revealed by the two-way repeated-measures ANOVA analyses, showing a significant interaction between treatment (30% sucrose exposure) and time (recovery) (grip strength test *p* = 0.005, corridor test *p* = 0.002). As for the HFD in IR-HG mice, IR by 30% sucrose in the drinking water in IR-NG mice led to no differences in cerebral blood flow reduction during MCAO and a non-statistically significant trend towards increased striatal neuronal loss versus healthy controls (*p* = 0.0514) (Figure 2r). Figure 2s shows a representative NeuN staining of striatal stroke induced by tMCAO.

We also examined the metabolic parameters 4 weeks after stroke. Weight, glycemia and hyperinsulinemia normalized quickly in IR-HG mice, showing no significant differences versus control mice (Figure 3a–c). Insulin sensitivity also improved in IR-HG mice but the response to insulin remained diminished in IR-HG mice compared to the CTRL group, as detected by two-way repeated-measures ANOVA (diet effect: *p* = 0.0498) and the calculated area under the curve (Figure 3d,e). The weight also decreased in IR-NG mice, remaining similar to control mice (Figure 3f), while glycemia remained unaltered. Furthermore, hyperinsulinemia was also normalized entirely after stroke in IR-NG mice (Figure 3g). Similar to IR-HG, insulin sensitivity was improved but remained lower in IR-NG mice compared to CTRL, as detected by two-way repeated-measures ANOVA (drinking water effect. *p* < 0.0001) and the calculated area under the curve (Figure 3h,i).

Altogether, these results indicate that pre-stroke IR, even without concomitant hyperglycemia, significantly impairs post-stroke neurological recovery. The results also indicate that impaired recovery by IR is not the result of increased neuronal loss.

### 2.3. IR Significantly Increases Stroke-Induced Inflammation

To evaluate stroke-induced inflammation, we quantified Iba-1 immunoreactivity in the ipsilateral, stroke-damaged striatum versus contralateral and respective sham mice. Stroke led to significantly increased Iba-1 in the ipsilateral striatum in all mice (Figure 4a,c,e). However, this increase was significantly greater in IR-NG mice versus CTRL (Figure 4c), while a non-statistically significant trend towards increased Iba-1 was also observed in IR-HG mice versus CTRL (Figure 4a). We also quantified the subpopulation of activated microglia expressing CD68 in the ipsilateral striatum but found no significant differences between the groups (Figure 4b,d,f).

These results suggest an overall exacerbation of the neuroinflammatory response after both HFD (IR-HG mice) and 30% sucrose in drinking water (IR-NG mice), although, at 4 weeks after stroke, this response is probably in its descending phase (no effect in CD68^+^ cells). Our data indicate an association of decreased insulin sensitivity with increased neuroinflammation after stroke, even in the absence of hyperglycemia.

### 2.4. The Number of ChAT^+^ Interneurons Pre- and Post-Stroke Is Differently Regulated in IR-HG and IR-NG Mice

We investigated the association between decreased stroke recovery and potential changes in choline acetyl transferase^+^ (ChAT^+^) interneurons in IR-HG and IR-NG mice. These interneurons have been shown to play an important role in striatal function [38] and stroke recovery [39].

We recorded a significant increase in striatal ChAT^+^ interneurons in IR-HG sham mice versus healthy CTRL and a strong trend in the same direction between IR-NG and CTRL mice (*p* = 0.06) (Figure 5a,c). No differences were found when the contra- or ipsilateral hemispheres were compared with their respective shams in CTRL or IR-HG mice (Figure 5b,e). Interestingly, the number of ChAT^+^ interneurons was significantly decreased in the ipsilateral hemispheres of IR-NG compared to their respective shams (Figure 5b,e). Moreover, the number of ChAT^+^ interneurons in the ipsilateral striatum of IR-NG was significantly lower than in the correspondent contralateral hemisphere and the ipsilateral hemisphere of CTRL mice (Figure 5d,e).

### 2.5. The Pre-Stroke Normalization of IR Improves Functional Recovery in Middle-Aged Mice

To further demonstrate the specific role of decreased insulin sensitivity in stroke recovery in a preclinical setting of clinical relevance, we utilized middle-aged mice that spontaneously developed early IR. To verify early IR in middle-aged mice, we performed a head-to-head comparison of 10-month-old vs. 1-month-old mice. The comparison showed that middle-aged mice remained normoglycemic (Figure 6a), while the response to insulin was reduced, as detected by two-way repeated-measures ANOVA (aging effect: *p* < 0.0001) and the calculated area under the curve (Figure 6b,c). After early IR was verified, 9-month-old middle-aged mice were randomly assigned to receive Rosiglitazone (a peroxisome proliferator-activated receptor gamma (PPARγ) and an insulin-sensitizing agent) or vehicle treatment for 3 weeks. As expected, glycemia was not affected (Figure 6e), while Rosiglitazone did improve insulin sensitivity, as detected by two-way repeated-measures ANOVA (treatment effect: *p* < 0.0001) and the calculated area under the curve (Figure 6f,g). Moreover, in accordance with the literature, treatment with Rosiglitazone induced a small but significant increase in body weight (Figure 6d).

After the normalization of insulin sensitivity, Rosiglitazone treatment was withdrawn for 24 h before transient middle cerebral artery occlusion (tMCAO) to avoid potential acute neuroprotective effects. The stroke recovery was assessed as previously described by the grip strength test and corridor test. Rosiglitazone treatment significantly improved the recovery of grip strength (Figure 6h) and stroke-induced sensory lateralization versus the vehicle (Figure 6i). In fact, a two-way repeated-measures ANOVA analysis revealed a significant interaction between treatment (Rosiglitazone) and time (recovery) (grip strength test *p* = 0.0002, corridor test *p* = 0.006). In line with the results obtained in the IR-HG and IR-NG models (Figure 2o,r), this effect on recovery was not associated with changes in neuronal cell death (Figure 6j). No differences were recorded in ipsilateral striatal Iba-1 expression between IR-VH and IR-Rosi mice (Figure 6k).

Taken together, these results strongly support a causative role of IR in impaired stroke recovery in both prediabetes and T2D.

## 3. Discussion

The aim of this study was to determine whether pre-stroke IR affects post-stroke functional recovery. We also investigated some of the potential cellular mechanisms involved. We show that the induction of IR before stroke worsens post-stroke functional recovery and that this effect is independent of hyperglycemia. Moreover, we demonstrate that targeting IR pharmacologically counteracts this effect, supporting a causative role of IR in impaired stroke recovery. Finally, we show that the detrimental effect of IR on stroke recovery is associated with increased post-stroke neuroinflammation and impaired expression of ChAT+ interneurons.

Hyperglycemia is present in 30–40% of people with acute ischemic stroke, even without a history of diabetes [7,9]. However, randomized clinical trials failed to show a benefit from intensive glucose control acutely (up to 72 h) after stroke [12]. There are potential explanations for this outcome that need to be addressed in future studies. For instance, hyperglycemia at admission could be only a reflection of pre-stroke diabetes-induced cerebral pathology and therefore it should be treated for a longer time than 72 h, as also suggested in preclinical studies by our group [29,34]. Regardless, additional and/or alternative avenues to improve post-stroke functional recovery in T2D need to be investigated considering that the medical need in this area is dramatically rising and established clinical guidelines for improving post-stroke functional recovery in T2D are entirely lacking.

IR is a pathogenic condition that is present in both prediabetes and T2D and is involved in several processes that are detrimental to stroke recovery, such as impaired lipid metabolism and obesity, endothelial dysfunction, hypertension and atherosclerosis [40]. Furthermore, brain IR has been associated with brain dysfunction and neurodegenerative disorders [41]. A few clinical studies have recently investigated the potential association between IR and poor functional outcomes after stroke. For instance, IR has been associated with poor clinical outcomes [42] even after thrombolytic treatment [43,44]. In addition, two recent studies have shown that β-cell dysfunction is significantly associated with poor stroke prognosis [45,46]. Moreover, Jing and colleagues demonstrated that IR at 14 days after stroke was associated with poor outcomes, but not dependence, in non-diabetic stroke patients [47]. Similarly, Ago and colleagues have shown that IR (assessed 8.3 ± 7.8 days after stroke onset) was independently associated with poor functional outcomes in patients with and without T2D [21]. Finally, a recent study in non-diabetic stroke patients proved that IR was associated with poor stroke outcomes, but, after adjustment for stroke severity, this association lost significance [23]. Overall, these studies show that IR is associated with worse stroke outcomes and recovery in both T2D and non-diabetic people. However, these studies were mainly observational, and they did not prove whether IR is a causal factor of worsened stroke recovery or if it is rather part of, and/or induced by, the acute/subacute stroke morbidity. In fact, in all these studies, IR was assessed at admission or later (even weeks) after stroke onset. This is a key question that needs to be answered to understand the suitability for potential therapeutics aimed to improve stroke recovery, based on the targeting of IR. Here, we demonstrated that the selective induction of early IR, without hyperglycemia as a confounding factor, worsens stroke recovery and that the pharmacological treatment of IR with the PPARγ agonist Rosiglitazone counteracts this effect, strongly supporting IR as a causative factor at the basis of impaired stroke recovery.

In Study 1 and 2, we induced early IR with or without hyperglycemia in adult mice to determine a potential causative role of IR in impaired stroke recovery. We are aware that stroke is highly prevalent in the middle-aged population and that aging also significantly impacts metabolism. However, to avoid aging as a confounding factor and to be able to determine the specific role of IR in impaired recovery, we specifically chose to use younger animals (5 months old at stroke induction) in Study 1 and 2. We demonstrated that the sucrose-supplemented drinking water selectively induced early IR, without confounding factors such as hyperglycemia, obesity or aging. The observation that, in this model, IR induction was sufficient to impair stroke recovery could have very important clinical implications. It is known that increased sugar consumption disrupts immune-mediated protection from metabolic syndrome [48] and is increasingly considered as a contributor to the consequent cardiometabolic disease [49]. However, we show here, for the first time, that excess sugar consumption also impairs functional recovery after stroke. This finding might have widespread implications for human lifestyle and nutrition guidelines.

To study the efficacy of pre-stroke IR normalization for post-stroke recovery in a more clinically relevant murine model, we took advantage of middle-aged mice that spontaneously developed early IR/prediabetes, also in this case without hyperglycemia. Indeed, the high prevalence of IR/prediabetes in the elderly is well known [19,20]. Aging is also the strongest non-modifiable risk factor for stroke [50] and aged people have the highest stroke mortality and most severe and long-lasting stroke sequelae [22]. Taking into account these strong associations and the fact that the number of elderly people is expected to double by 2050 according to the United Nations (https://www.un.org/en/development/desa/population/publications/pdf/ageing/WorldPopulationAgeing2019-Highlights.pdf accessed on 20 December 2022), the results of our study, showing that the PPARg agonist and insulin sensitizer Rosiglitazone enhances stroke recovery, encourage the launch of new clinical studies aimed at understanding whether preventive pharmacological interventions to normalize IR in aged people can improve post-stroke functional recovery. Remarkably, elderly people with IR do not routinely receive any active treatment to normalize insulin sensitivity. Therefore, a positive outcome from these studies will have novel clinical implications, which could benefit an enormous group of people. The PPARγ agonist Pioglitazone was recently associated with lower risk of recurrent stroke in patients with IR, prediabetes and T2D with a history of stroke or transient ischemic attack ([51]) (reviewed in [14,52]). Moreover, a preclinical study using pre-stroke Pioglitazone administration to *db/db* mice ameliorated neurological outcomes 24 h after tMCAO ([53]). However, no study has investigated whether the pre-treatment of IR (in prediabetes or T2D) by PPARγ agonists can improve post-stroke functional recovery. Preclinical research using PPARγ agonists has shown that these drugs also reduce stroke-induced brain damage in animal models of stroke, effects that could be behind the improved stroke recovery (reviewed in [14,54]). However, these effects were achieved when the drug was administered acutely around stroke onset, suggesting that acute neuroprotection, rather than effects by PPARγ agonists through the regulation of IR, was the main mechanism behind these effects. To avoid the possible confounding factor of neuroprotection in our studies, Rosiglitazone was administered for 3 weeks before stroke, but the treatment was withdrawn 24 h before inducing stroke, with the goal to avoid the concurrency of these acute neuroprotective effects. By showing that Rosiglitazone improves post-stroke functional recovery, our results are particularly relevant because they strongly support the hypothesis that this treatment improves stroke recovery independently of acute neuroprotection. Indeed, these data were indirectly confirmed in Studies 1–2, where decreased post-stroke recovery in IR-HG and IR-NG mice occurred without significant effects on neuronal cell death.

Recovering from stroke is a complex process to restore impaired sensorimotor function but also to overcome, among others, speech and emotional troubles, depression and attention and memory problems. Therefore, IR can very likely affect stroke recovery through pleiotropic mechanisms, both peripheral and central, which are too extensive to be addressed in one study. To start gaining new knowledge in this field, we have investigated the potential role of neuroinflammation, since the interplay between IR and increased neuroinflammation is well known [55]. We did not detect differences in Iba-1^+^ microglia in sham-operated animals, suggesting that pre-stroke neuroinflammation was likely not involved in the detrimental effects of decreased insulin sensitivity on stroke recovery. However, we recorded a strong trend towards an increase in Iba-1^+^ microglia in the stroke-damaged striatum of IR-HG mice versus healthy CTRL mice in Study 1. This effect was significant when comparing the striatum of IR-NG mice versus CTRL mice, suggesting that decreased insulin sensitivity exacerbates the neuroinflammatory response after stroke. However, when assessing ipsilateral Iba-1 expression, no difference between the IR-VH and IR-Rosi groups was found and therefore we cannot confirm at this stage that impaired inflammation plays a direct role in the worsening of stroke recovery by IR. There are weaknesses in the neuroinflammation assessments of our studies that need to be acknowledged. First, Iba-1 was quantified at only one time point (4 weeks after stroke). This likely was too late, since the post-stroke neuroinflammatory response was probably already in its descending phase, as also suggested by the CD68 data, showing that the number of these cells was unaffected at this time point. Secondly, we recorded trends towards increased neuronal loss in both the IR-HG and IR-NG groups versus healthy CTRL mice that could in part explain the increased neuroinflammation. Therefore, future studies will be needed to characterize the microglial response more thoroughly at additional time points during the post-stroke recovery phase, by also using additional markers of neuroinflammation. Additionally, the effect of IR on other cell types involved in the neuroinflammatory process will need to be investigated. For instance, the interplay between pericytes and macrophages has been recently shown to contribute to brain repair and recovery [56]. Moreover, pericytes affect oligodendrogenesis and astrogliosis [57]. Therefore, it could be of great interest to investigate whether IR impairs these processes of repair.

To investigate mechanisms of neuroplasticity, we focused our studies on striatal cholinergic ChAT^+^ interneurons, since these cells coordinate the firing of medium spiny neurons that in turn regulate motor output [38] and also regulate stroke recovery [39]. Other interneurons, such as those positive for Parvalbumin and Somatostatin, have been recently investigated in other studies from our group [17,25,26,29,34]. We recorded an overall increased number of ChAT^+^ interneurons in IR-HG and IR-NG sham-operated mice versus healthy CTRL mice, likely indicating an upregulation of ChAT expression, making more cells detectable by IHC. Although speculative, these data suggest an abnormal condition of this subgroup of striatal interneurons under both IR/hyperglycemic and IR/normoglycemic conditions, despite “normal” striatal motor function in the absence of stroke. This abnormality of cholinergic interneurons could, however, manifest functionally after stroke, as indicated by the impaired grip strength recovery and sensory lateralization in the corridor test. We also showed that ChAT^+^ interneurons were decreased significantly after stroke in the ipsilateral stroke-damaged striatum of IR-NG mice versus both their correspondent contralateral hemispheres and CTRL mice. However, we did not record a similar effect in IR-HG. Since both IR-NG and IR-HG mice are IR, the results could suggest that ChAT^+^ interneuron-mediated neuroplasticity is not involved in post-stroke recovery. On the other hand, our assessments were performed at only one time point after stroke and this represents a limitation of this study, and new experiments (including functional studies) will need to be performed in the future. However, these results are intriguing because they are not supported by an overall effect of IR on neuronal death. Therefore, they could serve as a basis to investigate further the potential neuroplastic role of ChAT^+^ interneurons after stroke, in the absence of stroke-induced brain damage.

Insulin locally regulates several brain functions, including synaptic plasticity and dendritic spine formation, neurotransmitter turnover, vascular function and lipid metabolism [58]. Thus, impaired insulin production following peripheral IR could directly affect all these functions, thereby impairing stroke recovery. On the other hand, “brain IR” itself (failure of brain cells to respond to insulin) could also be involved [58,59]. Neurons are the cells that have been mostly characterized in relation to the insulin-regulated functions listed above [59]. However, studies investigating the potential neuronal impairment after stroke in IR mice are lacking and will be needed in the future. An impaired response to insulin from other brain cells could also affect stroke recovery. For instance, the insulin effects on vascular smooth muscle cells are important for transporting and communicating nutrients, cytokines, hormones and other signaling molecules [60]. Impairments of insulin action on the vessels are major contributors to macro- and microvascular diseases [60] and may likely affect negatively post-stroke recovery. Indeed, the impairment of the IRS1/2/PI3K/Akt pathway (triggering insulin signaling) leads to the loss of pro-angiogenic, anti-oxidative and anti-inflammatory actions in genetically obese Zucker rats [61] and studies as such should also be performed in the brain after stroke.

Astrocytes are brain cells that play an important but complex role in stroke recovery in T2D [62]. For instance, insulin signaling in astrocytes modulates synaptic plasticity at dopaminergic axonal terminals [63]. Moreover, insulin signaling in hypothalamic astrocytes controls CNS glucose sensing and systemic glucose metabolism via the regulation of glucose uptake across the BBB [64]. In relation to stroke recovery, we also recently showed that HFD feeding results in reduced astrocyte reactivity in association with impaired recovery [17]. It is therefore plausible that impaired insulin signaling in astrocytes plays an important role in decreased recovery. Interestingly, a recent work by Fernandez and colleagues showed that insulin receptors in astrocytes participate in neurovascular coupling by modulating glucose uptake and angiogenesis [65]. Very little is known about insulin action on oligodendrocytes, but a recent clinical study indicates that insulin and IR influence white matter myelination [66]. This finding is very important since white matter myelination is a key process in stroke recovery, thus motivating further studies addressing the effects of insulin and IR on oligodendrocytes after stroke.

In summary, understanding the effects of impaired metabolism and IR on stroke recovery is a research field in its infancy. Therefore, future studies will be needed to identify the individual cells and mechanisms involved, both in the periphery and in the CNS.

## 4. Materials and Methods

### 4.1. Sample Size Calculation

Group sizes were determined based on a ≈20% effect size between groups in functional recovery with α = 0.05, statistical power of 90%. Standard deviation used in sample size calculation was obtained from pilot experiments. The analyses suggested the sample size of minimum n = 5 per group. However, after taking into consideration the success rate of stroke surgery, mortality and likelihood of statistical outliers, the experimental groups were set at n = 7–10 each.

### 4.2. Animals

In total, 80 male C57BL/6JRj mice (Janvier Labs, Le Genest-Saint-Isle, France) were used in this study. Mice were housed in environmentally controlled conditions (22 ± 0.5 °C, 12/12 h light/dark cycle with ad libitum access to food and water). The mice were housed under pathogen-free conditions in type III size, individually ventilated cages with wood chip bedding and nest material.

### 4.3. Experimental Design

#### 4.3.1. Study 1 (Pre-Stroke IR and Fasting Hyperglycemia Were Induced by Chronic High-Fat Diet (HFD) Feeding)

Thirty male mice were used. From 4 weeks of age, the mice were fed for 4 months with either standard laboratory chow (cat. 2918, 18% kcal from fat, 58% kcal from carbohydrates, 24% kcal from protein, Envigo, Indianapolis, IN, USA; n = 15, CTRL group) or HFD (cat. E15126-34, 54% kcal from fat, 29% kcal from carbohydrates, 17% kcal from protein, Ssniff, Soest, Germany; n = 15, IR-hyperglycemia (IR-HG) group). Obesity/T2D development in the IR-HG group was confirmed by a body weight increase beyond 20%, fasting glucose levels over 7 mmol/L, hyperinsulinemia and decreased insulin sensitivity (measured by insulin tolerance test (ITT)). The mice were then subjected to tMCAO (n = 10 CTRL, n = 10 IR-HG) or sham surgery (n = 5 CTRL, n = 5 IR-HG). After tMCAO, the HFD was substituted with SD in the IR-HG group, to reflect the clinical setting of a balanced post-stroke diet. See Figure 1a for the experimental design.

One mouse was removed from the IR-HG group (euthanized) shortly after tMCAO surgery, since the condition reached a humane endpoint. The final number of animals used in this study was as follows: CTRL group (n = 10) and IR-HG group (n = 9). The forelimb sensorimotor function (forelimb grip test) and lateralized sensorimotor integration (corridor test) were then measured weekly or at 1, 2 and 3 weeks after stroke, respectively. The mice were sacrificed at 4 weeks after stroke (time when the CTRL mice had fully recovered) and brains were collected for histology. See Figure 1a for the experimental design.

#### 4.3.2. Study 2 (Pre-Stroke IR, without Hyperglycemia, Was Induced by High Content of Sugar in the Drinking Water)

Thirty male mice were used. From 4 weeks of age, 15 mice were exposed to 30% sucrose in drinking water for 4 months to induce early IR but not fasting hyperglycemia (IR-normoglycemic (IR-NG) group). In the IR-NG group, decreased insulin sensitivity was verified as for Study 1. Mice drinking regular water were used as controls (n = 15, CTRL group). The mice were then subjected to tMCAO or sham surgery (n = 5 CTRL, n = 5 IR-NG), as for Study 1. After tMCAO, sucrose was withdrawn. See Figure 1b for the experimental design.

Three mice from the IR-NG group and two mice from the CTRL group were euthanized shortly after surgery, since their condition reached a humane endpoint.

The final number of animals used in this study was as follows: CTRL group (n = 8) and IR-NG group (n = 7). The upper-limb grip strength and lateralized sensorimotor integration were assessed as for Study 1. The mice were then sacrificed, and the brains were collected for histology. See Figure 1b for the experimental design.

We are aware that in order to test the hypothesis behind Studies 1–2, one study including 3 groups (CTRL, IR-HG, IR-NG) at the same time would have been ideal. Due to the availability of the mice, two separate studies have been instead conducted. However, age-matched controls have been used to allow for proper comparisons.

#### 4.3.3. Study 3 (Pre-Stroke IR Was Induced by 10 Months of Aging and Normalized by Rosiglitazone Treatment)

Twenty male mice were used. To study early IR/prediabetes in a clinically relevant manner, 10-month-old (middle-aged) mice were used (n = 15) since, at this age, early IR is spontaneously developed. Decreased insulin sensitivity was verified by ITT in a head-to-head comparison with 1-month-old young mice (n = 5). Afterwards, the IR mice were randomly divided into two groups: 7 mice treated with vehicle (IR-VH group), and 8 mice treated with Rosiglitazone for 3 weeks (10 mg/kg/day perioral; IR-Rosi group). After the treatment, the mice were subjected to tMCAO. See Figure 1c for the experimental design.

One mouse from the IR-VH group and one mouse from the IR-Rosi group were euthanized shortly after surgery, since the condition reached a humane endpoint. The final number of animals used in this study was as follows: IR-VH group (n = 6) and IR-Rosi group (n = 7). The same behavioral tests to assess stroke recovery were performed as for Studies 1–2, and the mice were then sacrificed and the brains were collected for histology at 5 weeks after stroke. See Figure 1c for the experimental design.

### 4.4. Transient Middle Cerebral Artery Occlusion

Stroke was induced by transient middle cerebral artery occlusion (tMCAO) using the intraluminal filament technique, as previously described [35]. Briefly, mice were anesthetized by inhalation of 3% isoflurane. Anesthesia was maintained by 1.5% isoflurane throughout the surgery. Body temperature was maintained at 37–38 °C using a heated pad with feedback from a thermometer. Left external (ECA) and internal (ICA) carotid arteries were exposed and a 7–0 silicone-coated monofilament (total diameter 0.17–0.18 mm) was inserted into the ICA until it blocked the origin of the MCA. After 30 min, the occluding filament was removed. In Studies 1 and 2, cerebral blood flow in the vicinity of MCA was monitored by a Laser Doppler Blood Flow Monitor (Moor Instruments Ltd., Axminster, UK) to evaluate whether the potential differences in stroke outcome between the groups were associated with differences in blood flow/MCA occlusion. Stroke induction was considered unsuccessful when the occluding filament could not be advanced within the internal carotid artery beyond 7–8 mm from the carotid bifurcation, or mice lacked symptoms of neurological impairment based on the neurological severity score [36]. All mice were given analgesic (Carprofen, 5 mg/kg) and soft food after the surgery.

### 4.5. Fasting Glycemia and ITT

Fasting glycemia was measured after overnight fasting using a glucometer and blood from a tail tip puncture. For ITT, mice were fasted for 3 h and injected intraperitoneally with human insulin (0.5 unit/kg) in saline. Blood glucose was measured before insulin injection (baseline) and at 15, 30, 45, 60, 75 and 90 min after injection. For analyses, the area under the curve (AUC) was computed.

### 4.6. Assessment of Sensorimotor Function

#### 4.6.1. Forelimb Grip Strength

The forelimb grip strength [25,34] was measured by using a grip strength meter (Harvard Apparatus, Holliston, MA, USA) before, at 3 days and at 1–4 weeks after tMCAO. Briefly, mice were firmly held by the body and allowed to grasp the grid with the right forepaw. Mice were then dragged backwards until the grip was broken. Ten trials were performed, and the highest value was recorded, as described previously [25,34].

#### 4.6.2. The Corridor Test

The corridor test was performed in a 50-cm-long, 4-cm-wide and 15-cm-high Plexiglas corridor to assess the lateralized sensorimotor integration, as described by Wattananit and colleagues [67]. Briefly, the mice were first habituated to the corridor for 5 min 2 days before the test. On the day of the testing, mice were fasted for 6 h. Mice were habituated for 2 min in an empty corridor and then immediately transferred to an identical corridor with 10 pots on each side, each containing a flavored treat. The number of explorations made by the mouse at the ipsilateral or contralateral to injury side was counted for 5 min. The data are presented as the ratio of ipsilateral/contralateral explorations. All behavioral tests were performed blinded to the experimental groups, although this was not always possible due to obvious weight differences in Study 1.

### 4.7. Serum and Tissue Collection

The mice were deeply anesthetized by an overdose of sodium pentobarbital. Then, blood was collected via cardiac puncture and left to clot at RT for 20 min. Hereafter, blood samples were centrifuged for 15 min using a benchtop centrifuge at 2000× *g*. Serum was collected and stored at −80 °C. After cardiac puncture, mice were transcardially perfused with saline, followed by 4% ice-cold paraformaldehyde. Brains were dissected out and, after overnight post-fixation, transferred to a solution of phosphate-buffered saline (PBS) with 25% sucrose until they sank. The brains were then cut into 30-μm-thick coronal sections using a sliding microtome and stored in anti-freeze solution at −20 °C.

### 4.8. Immunohistochemistry (IHC)

The brain tissue staining was performed by the free-floating method. Briefly, brain sections were washed in PBS and then incubated with PBS containing 3% H_2_O_2_ and 10% methanol for 20 min at RT to quench endogenous peroxidases. The sections were then blocked in PBS containing 3–5% normal serum and 0.25% Triton-X-100 for 1h (at RT), and incubated overnight in primary antibody solution at 4 °C.

The following primary antibodies were used: mouse anti-NeuN (1:500 dilution, #MAB377, Millipore, Burlington, MA, USA; RRID:AB_2298772), a neuronal marker; goat anti-Iba-1 (1:1000 dilution, #ab5076, Abcam, Cambridge, UK; RRID:AB_2224402), a marker for microglia; rabbit anti-CD68 (1:2000 dilution, #ab125212, Abcam; RRID:AB_10975465), a marker for phagocytic microglia and macrophages; goat anti-ChAT (1:1500 dilution, #AB144P, Millipore; RRID:AB_2313845), a marker for cholinergic interneurons.

After overnight incubation with the primary antibody solution, sections were washed and incubated for 2 h at RT with the secondary antibody. The following secondary antibodies were used: biotinylated horse anti-mouse (1:200 dilution, #BA-2000, Vector Laboratories; Newark, CA, USA; RRID:AB_2313581); biotinylated horse anti-goat (1:200 dilution, #BA-9500, Vector Laboratories; RRID:AB_2336123); biotinylated horse anti-rabbit (1:200 dilution, #BA-1100, Vector Laboratories; RRID:AB_2336201). Incubation with biotinylated secondary antibody was followed by incubation with avidin–biotin complex (1:200 dilution for both reagent A and B, Vectastain Elite ABC kit, Vector Laboratories) for 1 h at RT, followed by development with DAB.

### 4.9. Quantitative Microscopy and Image Analysis

#### 4.9.1. Assessment of Stroke-Induced Brain Damage by Quantifying NeuN^+^ Surviving Neurons

NeuN^+^ cells were counted using a computerized setup for stereology, driven by the StereoInvestigator software (MBF Bioscience, Williston, VT, USA). NeuN staining is a consistent method for quantifying neural damage because it exclusively stains neurons. Therefore, it is reliable to evaluate neuronal loss even several weeks post-stroke, unlike ubiquitous cell markers, such as 3,5-triphenyltetrazolium chloride, which are accurate only within a few days after the injury because of later inflammatory cell infiltration and glial scar formation. The number of neurons was quantified using the optical fractionator method [68,69]. Briefly, brain sections were displayed live on the computer monitor and the striatum was delineated at 1.25× magnification. Quantifications were performed using a dry 63× lens. Eight evenly spaced serial sections throughout the entire striatum were included. Random sampling was carried out using the counting frame, which systematically was moved at predefined intervals so that ∼300 immunoreactive cells were counted. The total number of cells was estimated according to the optical fractionator formula [68,69] and the percent decrease versus undamaged contralateral striatum was calculated.

#### 4.9.2. Assessment of Neuroinflammation

The Fiji open-source image analysis software was used to evaluate Iba-1 and CD68 immunoreactivity [70]. Briefly, images of Iba-1 staining in the striatum were acquired at 20× magnification using the Olympus BX40 microscope. Images were then converted into grayscale (8-bit) mode and thresholded. The lowest Iba-1 immunoreactivity in the CTRL sham group of each study was used as a baseline to determine the threshold. For each hemisphere, 3 images containing >90% of the striatum were analyzed, resulting in a total of 9 pictures analyzed per hemisphere per animal. The Iba-1^+^ area was measured and expressed as a percentage of the total area. CD68 images were acquired at 4× magnification. Then, the striatum was delineated and converted into grayscale (8-bit) mode. The lowest CD68 immunoreactivity in the CTRL sham group of each study was then used as a baseline to determine the threshold. After thresholding, the CD68^+^ area was determined and expressed as a percentage of the total striatal area.

#### 4.9.3. Analysis of Cholinergic Interneurons

Manual counting of ChAT^+^ cholinergic interneurons in the striatum was performed on three consecutive sections using the Olympus BX40 microscope. The first section was chosen based on the anatomical location along the rostral–caudal axis (approximately 1 mm from Bregma). The second and third sections were 300 and 600 μm caudal from the first section, respectively. ChAT^+^ cells were counted manually by experimenters blinded to experimental groups. Values were then normalized on the respective sham group and expressed as a fold increase over sham.

### 4.10. Insulin and Plasminogen Activator Inhibitor-1 (PAI-1) Enzyme-Linked Immunosorbent Assays (ELISA)

The levels of insulin and PAI-1 were quantified in mouse serum samples using ELISA kits (CrystalChem, 90,080 for insulin and Abcam, ab197752 for PAI-1) and utilized according to the manufacturer’s recommendations.

### 4.11. Data and Statistical Analysis

The data were checked for statistical outliers by using the ROUT method, and for normality by using the Shapiro–Wilk normality test, to decide whether to perform parametric or non-parametric tests.

Parametric tests: For pre- and post-stroke metabolic parameters, CD68 and NeuN analysis, Welch’s *t*-test was used. For the behavioral tests, two-way repeated-measures ANOVA with Geisser–Greenhouse’s correction followed by Dunnett T3 was used. For Iba-1 and ChAT analysis, two-way ANOVA followed by the two-stage linear step-up procedure of Benjamini, Krieger and Yekutieli was used. All data were analyzed by GraphPad Prism Version 9.0. Data are expressed as mean ± SD. A *p*-value less than 0.05 was considered statistically significant.

## 5. Conclusions

Neurological deficits are more severe and persist longer (even permanently) in stroke survivors with T2D and IR/prediabetes [42,47]. This clinical problem, in combination with the increased predicted number of T2D patients [71] and elderly people, but also with the increased stroke risk in both groups, will dramatically increase the number of stroke patients in need of care. We have employed validated animal models to support the causal role of IR to impair stroke recovery in T2D and aging. This is a very challenging goal to prove in the clinical setting, where it is essentially impossible to regulate IR in specific time windows around stroke onset and without affecting glycemia, a confounding factor, since hyperglycemia worsens stroke recovery. Importantly, we also showed that a diet supplemented with high sucrose content in the drinking water impaired insulin sensitivity, providing potentially important clinical implications for the detrimental role of high sucrose on stroke recovery. Finally, we showed that impaired stroke recovery can be treated through the normalization of IR. From a clinical perspective, these findings are very interesting because they might provide the opportunity to clinically investigate the potential efficacy of a specific and prophylactic targeting of IR to limit long-term sequelae in two of the largest groups of people at risk of suffering from stroke.

## Figures and Tables

**Figure 1 ijms-24-03989-f001:**
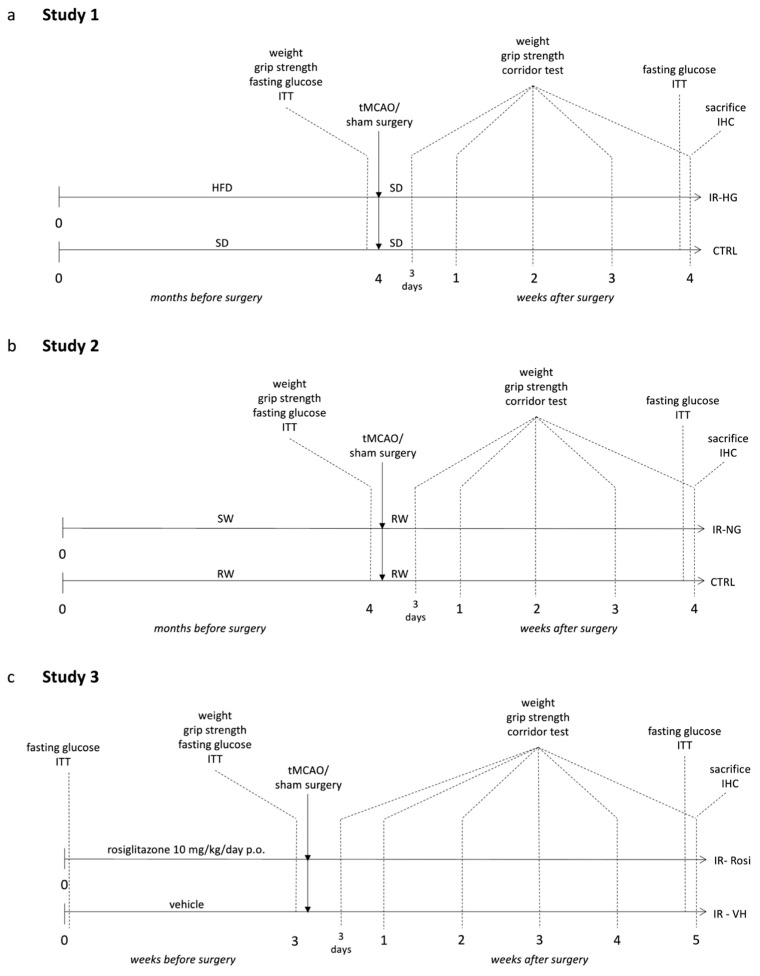
Experimental design. (**a**) Experimental design of Study 1: Pre-stroke IR and fasting hyperglycemia (IR-HG) were induced by chronic high-fat diet feeding. (**b**) Experimental design of Study 2: Pre-stroke IR, without hyperglycemia (IR-NG), was induced by 30% sucrose in the drinking water. (**c**) Experimental design of Study 3: Pre-stroke IR was induced by 10-month-aging and normalized by Rosiglitazone, respectively. SD = standard diet, HFD = high-fat diet, ITT = insulin tolerance test, IHC = immunohistochemistry, tMCAO = transient middle cerebral artery occlusion, SW = sucrose-supplemented water, RW = regular water.

**Figure 2 ijms-24-03989-f002:**
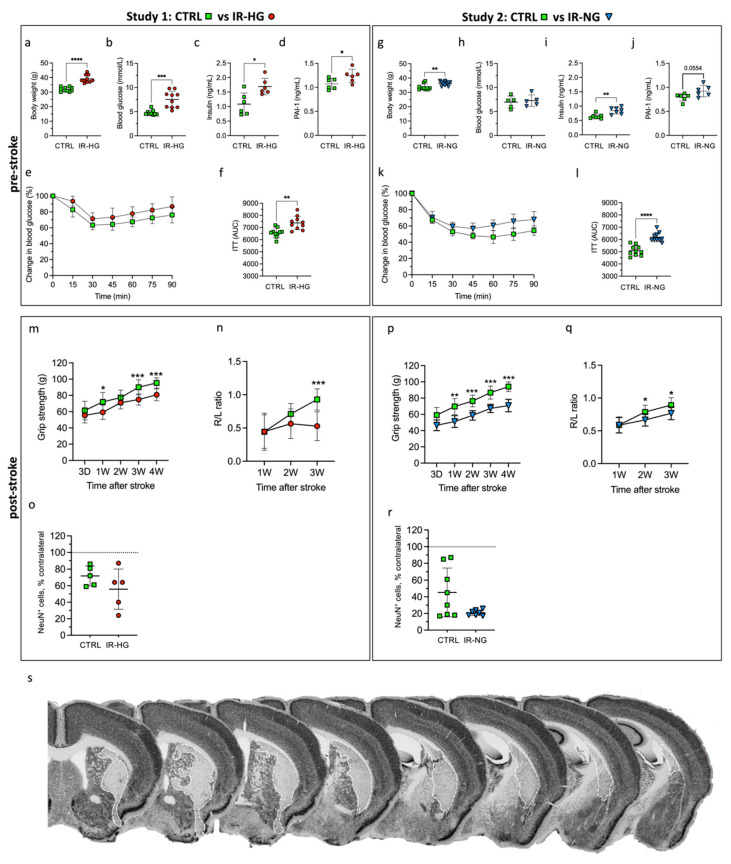
Effect of 4 months of HFD or exposure to 30% sucrose-supplemented drinking water on metabolic parameters and functional recovery after tMCAO. (**a**–**f**) Effect of 4 months of HFD feeding on body weight (**a**), fasting glucose (**b**), plasma insulin (**c**), plasma plasminogen activator inhibitor-1 (PAI-1) (**d**) and insulin sensitivity, shown as plotted curve (**e**) and area under the curve (**f**). (**g**–**l**) Effect of 4 months of 30% sucrose-supplemented drinking water on body weight (**g**), fasting glucose (**h**), plasma insulin (**i**), PAI-1 (**j**) and insulin sensitivity, shown as plotted curve (**k**) and area under the curve (**l**). (**m**–**r**) Forepaw grip strength (**m**) and corridor task (**n**) after stroke, Study 1. Forepaw grip strength (**p**) and corridor task (**q**) after stroke, Study 2. Percentage of remaining NeuN^+^ cells in the ipsilateral stroke-damaged striatum for Study 1 (**o**) and Study 2 (**r**). Representative images of NeuN staining (**s**). The white dotted lines on the images indicate the stroke area. Data are presented as mean ± SD. Statistical significance was calculated using Welch’s *t*-test (**a**–**d**,**f**–**j**,**l**,**o**,**r**) or two-way repeated-measures ANOVA followed by Benjamini, Krieger and Yekutieli multiple-comparisons test (**m**,**n**,**p**,**q**). Results were considered significant if *p* < 0.05. * denotes *p* < 0.05, ** denotes *p* < 0.01, *** denotes *p* < 0.001 and **** denotes *p* < 0.0001. Group sizes: (**a**,**b**,**e**,**f**,**m**,**n**): CTRL n = 10, IR-HG n = 9. (**c**,**d**,**o**): CTRL n = 5, IR-HG n = 5. G-l, (**p**–**r**): CTRL n = 8, IR-NG n = 7.

**Figure 3 ijms-24-03989-f003:**
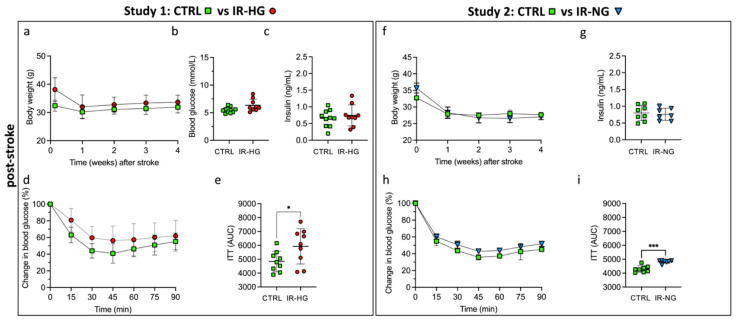
Body weight and metabolic parameters after tMCAO. (**a**–**e**): Body weight (**a**), fasting glucose levels (**b**), plasma insulin levels (**c**) and insulin sensitivity, shown as plotted curve (**d**) and area under the curve (**e**), 4 weeks after tMCAO in control (CTRL) vs. Type 2 diabetic (IR-HG) mice. (**f**–**i**): Body weight (**f**), plasma insulin levels (**g**) and insulin sensitivity, shown as plotted curve (**h**) and area under the curve (**i**), 4 weeks after tMCAO in control (CTRL) vs. insulin-resistant normoglycemic (IR-NG) mice. Data are presented as mean ± SD. Statistical significance was calculated using Welch’s *t*-test (**b**,**c**,**e**,**g**,**i**) or two-way repeated-measures ANOVA (**a**,**d**,**f**,**h**). Results were considered significant if *p* < 0.05. * denotes *p* < 0.05 and *** denotes *p* < 0.001. Group sizes: (**a**–**e**): CTRL n = 10, IR-HG n = 9. (**f**–**i**): CTRL n = 8, IR-NG n = 7.

**Figure 4 ijms-24-03989-f004:**
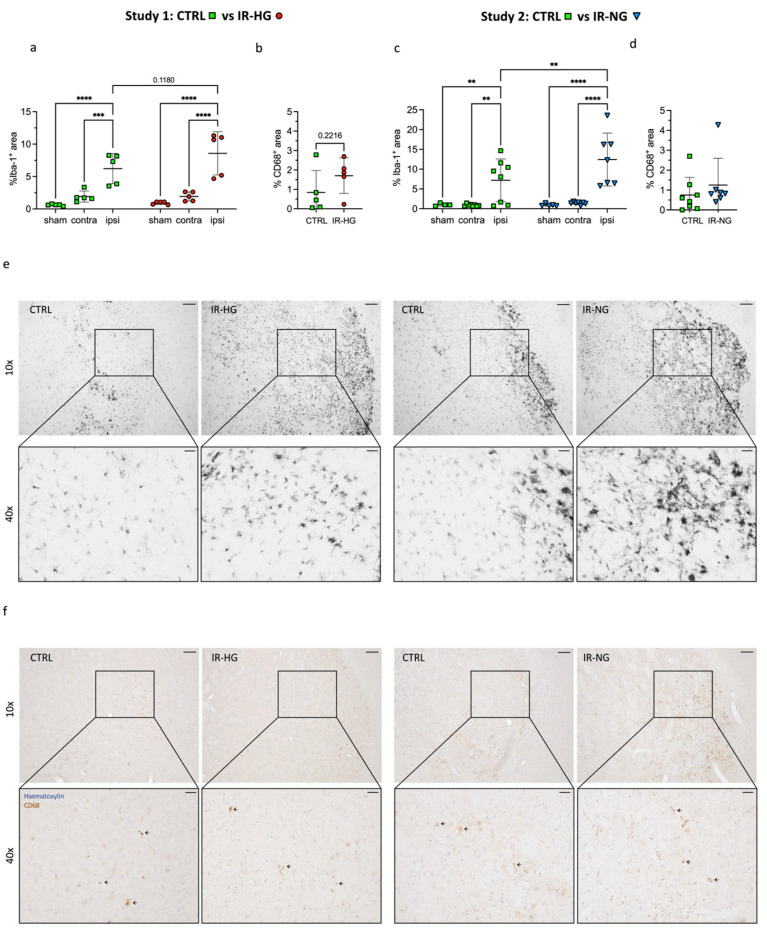
IR increases stroke-induced ipsilateral neuroinflammation. (**a**,**b**) Iba-1 (**a**) and CD68 (**b**) expression in striatum of control (CTRL) vs. Type 2 diabetic (IR-HG) mice 4 weeks after tMCAO. (**c**,**d**) Iba-1 (**c**) and CD68 (**d**) expression in striatum of control (CTRL) vs. insulin-resistant normoglycemic (IR-NG) mice 4 weeks after tMCAO. (**e**,**f**) Representative images of Iba-1 (**e**) and CD68 (**f**) in ipsilateral striatum of IR-HG and IR-NG and their respective controls. Arrows in (**f**) (bottom row) indicate CD68^+^ cells. Images were taken at 10× (upper row), scale bar = 100 μm, and 40× (bottom row), scale bar = 20 μm. Statistical significance was calculated using one-way ANOVA with Tukey’s post hoc test (**a**,**c**) or Welch’s *t*-test (**b**,**d**). Results were considered significant if *p* < 0.05. ** denotes *p* < 0.01, *** denotes *p* < 0.001 and **** denotes *p* < 0.0001. Group sizes: (**a**,**b**): CTRL sham n = 5, contralateral n = 5, ipsilateral n = 5. IR-HG sham n = 5, contralateral n = 5, ipsilateral n = 5. (**c**,**d**): CTRL sham n = 4, contralateral n = 8, ipsilateral n = 8. IR-NG sham n = 5, contralateral n = 7, ipsilateral n = 7.

**Figure 5 ijms-24-03989-f005:**
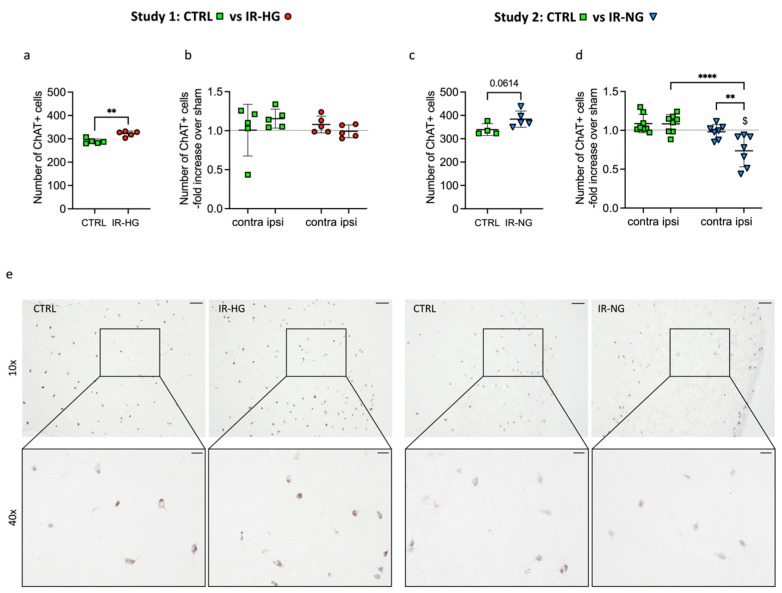
IR affects the number of ChAT^+^ interneurons both pre- and post-stroke. Number of ChAT^+^ cells in striatum of sham-operated control (CTRL) vs. Type 2 diabetic (IR-HG) mice (**a**) and CTRL vs. insulin resistant normoglycemic (IR-NG) mice (**c**). Number of ChAT^+^ cells in striatum 4 weeks after tMCAO normalized on sham of CTRL vs. IR-HG mice (**b**) and CTRL vs. IR-NG mice (**d**). Representative images of ChAT^+^ cells in striatum 4 weeks after tMCAO in CTRL vs. IR-HG mice and CTRL vs. IR-NG mice (**e**) at 10× (upper row), scale bar = 100 μm, and at 40× (bottom row), scale bar = 20 μm. Statistical significance was calculated using Welch’s *t*-test (**a**,**c**) or one-way ANOVA with Tukey’s *post hoc* test (**b**,**d**). Results were considered significant if *p* < 0.05. ** denotes *p* < 0.01, and **** denotes *p* < 0.0001. ^$^ depicts significant differences compared to the respective sham group. Group sizes: (**a**,**b**): CTRL sham n = 5, contralateral n = 5, ipsilateral n = 5. IR-HG sham n = 5, contralateral n = 5, ipsilateral n = 5. (**c**,**d**): CTRL sham n = 4, contralateral n = 8, ipsilateral n = 8. IR-NG sham n = 5, contralateral n = 7, ipsilateral n = 7.

**Figure 6 ijms-24-03989-f006:**
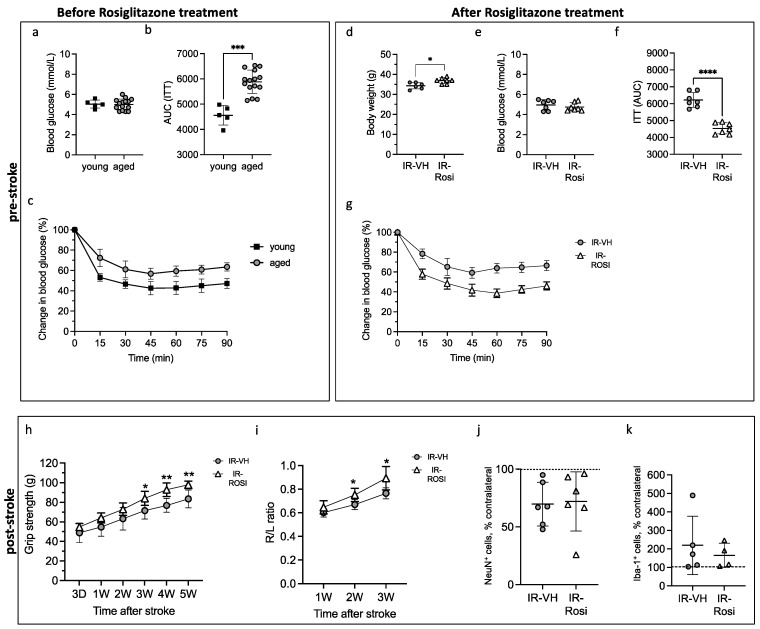
Rosiglitazone treatment normalizes aging-induced IR and significantly improves post-stroke functional recovery. (**a**–**c**) Fasting glucose levels (**a**) and insulin sensitivity shown as area under the curve (**b**) and plotted curve (**c**) of 1-month-old (young) vs. 9-month-old (aged) mice. (**d**–**f**) Body weight (**d**), fasting glucose levels (**e**) and insulin sensitivity shown as area under the curve (**f**) and plotted curve (**g**) of 10-month-old mice treated with vehicle (IR-VH) or Rosiglitazone (10 mg/kg/day) (IR-Rosi) for 3 weeks. Forepaw grip strength (**h**) and corridor task (**i**). Percentage of remaining NeuN^+^ cells in the ipsilateral stroke-damaged striatum after stroke (**j**). Percentage of Iba-1 expression in the ipsilateral stroke-damaged striatum after stroke (**k**). Data are presented as mean ± SD. Statistical significance was calculated using Welch’s *t*-test (**a**,**c**–**f**,**j**,**k**) or two-way repeated-measures ANOVA followed by Benjamini, Krieger and Yekutieli multiple-comparisons test (**h**,**i**). Results were considered significant if *p* < 0.05. * denotes *p* < 0.05, ** denotes *p* < 0.01, *** denotes *p* < 0.001 and **** denotes *p* < 0.0001. Group sizes: (**a**–**c**): young n = 5, aged n = 15. (**d**–**j**): IR-VH n = 6, IR-Rosi n = 7. (**k**): IR-VH n = 5, IR-Rosi n = 4.

## Data Availability

The data that support the findings of this study are available from the corresponding authors upon reasonable request.

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
