# Peer review of "The Pre-Stroke Induction and Normalization of Insulin Resistance Respectively Worsens and Improves Functional Recovery"

_ijms, 2023, doi:10.3390/ijms24043989_

Round 1
Reviewer 1 Report
In the present manuscript, Vercalsteren et al. examined the effect of insulin resistance (IR) on stroke recovery. The authors conducted their research on mice models. In the First study, pre-stroke IR and fasting hyperglycemia were induced in chronic high-fat diet-fed mice. In the second study pre-stroke IR, without hyperglycemia, was induced in 4 weeks old mice by giving high content of sugar in the drinking water, and in the Third study, Pre-stroke IR was induced in 10-month-old mice and regulated by Rosiglitazone treatment. In all three models, Stroke was induced by transient middle cerebral artery occlusion. The major outcome of the study is the pre-stroke induction of IR worsened neurological recovery. Impaired post-stroke neurological recovery can be treated with the normalization of IR. The study also revealed the possible involvement of neuroinflammation and cholinergic interneuron-mediated neuroplasticity in post-stroke recovery. Stroke recovery was assessed by sensorimotor tests. Neuronal survival, neuroinflammation, and neuroplasticity mediated by cholinergic interneurons were evaluated using immunohistochemistry quantitative microscopy.
· Stroke is most common in the middle-aged/old population. Metabolic parameters also change according to age. In study-1 and study-2, authors used 4 months old mice to induce stroke instead of middle-aged/aged mice. What's the rationale behind choosing young mice over middle-aged/old mice?
· Figure 4 (e) and Figure 4 (f ), especially Figure 4 (f ) look blurred. It could have been better if the authors can provide 10x and 40x, images instead of 4x and 20x
· In line 352 author mentioned “24 after tMCAO” is it “24 hrs after tMACO”?
· Check the spacing of line 339
Reviewer 2 Report
The focus of the study was on determining the effects of insulin resistance (IR), independent of hyperglycemia, on stroke recovery. The exploit an animal model: 1) high fat diet, HFD = IR with hyperglycemia or 2) regular diet with 30% sucrose water = IR without hyperglycemia.
In general, I thought the study was well written, with a solid rationale. The data and analysis were presented in a clear manner and appropriate stats were used. Therefore the results and conclusions were sufficiently rigorous. I have a few points that need to be addressed before publication.
Comments:
Abstract:
- I would not state that “neuroplasticity mediated by cholinergic interneurons” since there is no direct evidence in the paper to prove this, the metric is simply Ach neuron density. Therefore, stating “cholinergic interneuron density” on line 33-34 or “cholinergic neurons mediated neuroplasticity” on line 37 is more accurate.
Introduction:
- line 57 implies that controlling hyperglycemia has no clinical benefit (12), which probably isn’t what the authors meant to say. Reducing blood sugars has clinical benefit compared to not treating hyperglycemia at all. The study referenced shows that there was no difference between intensive vs standard glucose lowering methods.
Results:
- Figure 2, I don’t really see the point of showing grip strength measured as an AUC. I don’t even know what an AUC difference would even mean (other than its different). Therefore showing Grip strength in “g” is more than sufficient.
- The ischemic damage seems to be greater in study 2 than 1. Any insights into this?
- Figure 3D and H, was there not a significant main effect of group, at least in D? It is interesting that despite more normal insulin levels, the Insulin sensitivity is not completely normal in the IR+HG group, even after 4 weeks of normal diet. This may explain the delayed deficits you see in Fig 2m and o. This fits with some other animal work examining T1DM and delayed behavioural deficits after stroke.
- The authors should re-check the stats for Fig 4C stating that ipsi IBA levels are significantly different between control and IR-NG (doesn’t look like **p<0.01 difference)
- The authors should show the body weights for the Rosiglitazone treated mice, this is an important factor that could easily get overlooked if not presented in the figure 6.
Methods
- The authors should include their control and high fat diet composition in the methods. At least percent calories from each component fat, carbs, etc. They did not cite any relevant paper in case the composition was used from previous studies.
- The authors should mention the sex of mice in each study protocol.
- Page-17, Section 4.10: Serum collection method was missing in manuscript.
